



# Small-scale hydrological patterns in a Siberian permafrost ecosystem affected by drainage

Sandra Raab[1], Karel Castro-Morales[2], Anke Hildebrandt[3,4], Martin Heimann[1,5], Jorien Elisabeth Vonk[6], Nikita Zimov[7], Mathias Goeckede[1]

5   [1]Max Planck Institute for Biogeochemistry, Department of Biogeochemical Signals, Jena, 07745, Germany
[2]Friedrich-Schiller-University Jena, Institute of Biodiversity, Chair of Aquatic Geomicrobiology, Jena, 07743, Germany
[3]Helmholtz Centre for Environmental Research - UFZ, Department of Computational Hydrosystems, Leipzig, 04318, Germany
[4]Friedrich-Schiller-University Jena, Institute of Geoscience, Terrestrial Ecohydrology, Jena, 07749, Germany
[5]University of Helsinki, Institute for Atmospheric and Earth System Research (INAR), Helsinki, 00014, Finland
10   [6]Vrije Universiteit, Department of Earth Sciences, Faculty of Sciences, Amsterdam, 1081 HV, The Netherlands
[7]North-East Science Station, Pacific Institute for Geography, Far-Eastern Branch of Russian Academy of Science, Chersky, Republic of Sakha (Yakutia), 678830, Russia

*Correspondence to*: Sandra Raab (sraab@bgc-jena.mpg.de)



**Abstract.** Climate warming and associated accelerated permafrost thaw in the Arctic lead to a shift in landscape patterns, hydrologic conditions and release of carbon. In this context, the lateral transport of carbon, and shifts therein following thaw, remain poorly understood. Crucial hydrologic factors affecting the lateral distribution of carbon include e.g., the depth of the saturated zone above the ice table with respect to changes in water table and thaw depth, and the connectivity of water saturated zones. With changing landscape conditions due to rising temperatures, polygonal or flat floodplain Arctic tundra areas in various states of degradation are expected to become more common in the future, with associated changes in hydrologic conditions. This study centers in an experimental site near Chersky, Northeast Siberia, where a drainage ditch was constructed in 2004 reflecting landscape degradation features that result in drier soil conditions and channeled water flow. We characterized and compared the drained area (dry soil conditions) with an adjacent control area (wet soil conditions) with regard to water levels and thaw depths. We also identified the sources of water at the site via stable water isotope analysis. We found substantial spatiotemporal changes in the water conditions at the drained site: i) lower water tables resulting in drier soil conditions, ii) quicker water flow in drier areas, iii) larger saturation zones in wet areas, and iv) a higher proportion of permafrost melt water in the liquid phase towards the end of the growing season. These findings suggest a decreased lateral connectivity throughout the drained area. Shifts in hydraulic connectivity associated with a shift in vegetation abundance and water sources may impact carbon sources, sinks as well as its transport pathways. Identifying lateral transport patterns in areas with degrading permafrost are therefore of major relevance.

# 1 Introduction

Global warming can alter a variety of landscape processes, including the transformation and transport of water, carbon, and nutrients (AMAP, 2017; Walvoord and Kurylyk, 2016). In the Northern Hemisphere approximately 15 % of the land surface is underlain by permafrost (Obu, 2021). These areas represent a major reservoir for organic carbon storing up to 1300 PgC (Hugelius et al., 2014) and are susceptible to changing climate conditions (Treat et al., 2022; Zou et al., 2022). Especially in Siberia, large areas are covered by organic-rich loess soils that are highly vulnerable to global warming and therefore to organic decomposition (Zimov et al., 2006). The more permafrost thaws as a result of climate change (Lawrence et al., 2012; Osterkamp, 2007; Romanovsky et al., 2010), the more organic carbon becomes available for degradation and transport to the atmosphere (vertical release) or hydrosphere (lateral release) (Denfeld et al., 2013; Frey and McClelland, 2009; Schuur et al., 2015; Walvoord and Striegl, 2007; Vonk et al., 2015). The stability of this carbon reservoir therefore depends mainly on soil water status, temperature and vegetation community (Burke et al., 2013; Jorgenson et al., 2010, 2013; van der Kolk et al., 2016; Varner et al., 2021).

Vertical carbon release pathways in permafrost ecosystems were comprehensively studied over the past decade (Helbig et al., 2013; Zona et al., 2015). Particularly during summer, when the active layer develops as the seasonally thawed top section of the permafrost soil profile, fluctuations in carbon flux rates are often dominated by water availability (Kim, 2015; Kwon et al., 2016; McEwing et al., 2015; Zona et al., 2011). Permafrost represents an impermeable boundary forming the aquifer bottom



of the active layer groundwater (Lamontagne-Hallé et al., 2018; Woo, 2012). The deeper the soil profile thaws, the more water (from precipitation or flooding) infiltrates and moves through the soil towards lower areas following hydraulic gradients

(Hamm and Frampton, 2021) and towards inland water bodies enabling the redistribution of dissolved and particulate carbon from the active layer. These lateral water transport patterns are crucial to understand (Déry and Wood, 2005; Peterson et al., 2002) to quantify the total lateral carbon transport through an aquatic system. Carbon decomposition and transport rates also highly depend on water saturation of soils (dry vs. wet conditions). Therefore, the hydrosphere plays an important role in carbon redistribution and release at the lithosphere–hydrosphere interface (Denfeld et al., 2013; Goeckede et al., 2017;

Walvoord and Kurylyk, 2016; Woo et al., 2008). Recent publications show an increasing focus on understanding lateral groundwater fluxes combined with carbon export (Connolly et al., 2020; Ma et al., 2022; Mohammed et al., 2022).

Arctic mineral soils generally have low hydraulic conductivities that lead to low water and carbon transport rates within the area (Frampton et al., 2011; Zhang et al., 2000). Overlying organic layers in contrast are characterized by larger pore sizes and therefore higher permeabilities (Arnold and Ghezzehei, 2015; Boelter, 1969). When these organic layers are water saturated,

they have a higher potential to conduct water and facilitate lateral transport of carbon (O'Connor et al., 2019; Quinton et al., 2008). O'Connor et al. (2019) emphasized that groundwater flow is expected to be limited when water tables drop into the mineral layer. Therefore, it is still being debated whether deeper thawing leads to an enhanced or reduced groundwater flow (Evans and Ge, 2017; Walvoord and Kurylyk, 2016; Walvoord and Striegl, 2007). However, it has been shown that the possible vertical connectivity between suprapermafrost and subpermafrost groundwater can be enhanced due to increases in thaw depths

(Connolly et al., 2020; Kurylyk et al., 2014). Ultimately, all of this has an impact on the lateral transport within permafrost ecosystems: The lateral connectivity varies over the respective area depending on the seasonally driven water soil conditions during e.g., the spring freshet vs. late growing season.

Siberian floodplain areas are affected by widespread flooding during the spring freshet following snowmelt (Bröder et al., 2020; Mann et al., 2012; Raymond et al., 2007). With flooding, the water and carbon redistribution is facilitated compared to

later in the year when water levels are lower and transport occurs only via subsurface flow (Connon et al., 2014). O'Connor et al. (2019) underlined that the groundwater level location has a higher impact on suprapermafrost groundwater flow than the thaw depth location, in particular whether the saturated zone extends into the porous organic or less conductive mineral layer. Typical Arctic landscape patterns such as polygonal ice wedge formations, wetlands, thermokarst lakes, channels and ponds, are characterized by saturated soil water conditions during the growing season. A shift in landscape characteristics with drier

soil conditions is expected to become more frequent in the future, resulting in degradation of the polygonal tundra landscape (Frey and McClelland, 2009; Liljedahl et al., 2016). Drier conditions result in more channeled water transport pathways and aerobic soil conditions. This change in landscape patterns leads to a shift from grassy to shrubby vegetation community (Kwon et al., 2016; Sturm et al., 2001) and different decomposition patterns of carbon (Goeckede et al., 2017; Zona et al., 2011). Vonk et al. (2015) emphasized that physical, chemical and biological impacts of hydrological change can affect remobilization,

microbial transformation and carbon release from previously frozen soils.





Several studies have previously discussed the processes and drivers of water redistribution patterns in permafrost areas, both on large (mapping, remotely sensed data, modeling; e.g., Frey and McClelland, 2009; Koven et al., 2011; Rautio et al., 2011; Schuur et al., 2015) and small scales (e.g., Quinton et al., 2000; Walvoord and Kurylyk, 2016). However, the relation between wetness status and water flow velocities with regard to carbon distribution and transport remains understudied.
Microtopographic features (e.g., local elevations and depressions; O'Connor et al., 2019) with variations in water table and thaw depths, reveal different storage capacities and therefore the potential for different carbon decomposition or accumulation patterns which need to be integrated in future research.

In this study, we investigated how small-scale suprapermafrost groundwater distribution, potential flow paths and mechanisms are interlinked at a wet (control) and a dry (drained) permafrost site in northeast Siberia. We use several in situ approaches to 90 detect temporal changes and water redistribution patterns in hydrological features: small-scale water table depth patterns, composition of water stable isotopes ($\delta^{18}O$, $\delta D$) in surface water, suprapermafrost groundwater, permafrost ice and precipitation, and thaw depths measurements. We aimed to answer the following research questions: (i) How is the artificial drainage affecting the wet tussock tundra ecosystem in northeast Siberia? (ii) Which hydrological differences can be detected in wet and dry areas of this system? (iii) What are the changes that are induced by the drainage? (iv) Which relationships 95 between ecosystem structure and hydrological patterns can be observed?

## 2 Material and methods

### 2.1 Study site

The study site (centered at 68.75 °N, 161.33 °E) is located in the continuous permafrost zone on the floodplain of the Kolyma River in Northeast Siberia, Russia, close to the town of Chersky (Fig. 1). Located about 150 km south of the Arctic Ocean, the 100 landscape is characterized by frost polygons and ice wedge formations (Corradi et al., 2005). The study site is situated adjacent to the Ambolikha River, which enters the Pantheleika River and subsequently the Kolyma River (Castro-Morales et al., 2022). To simulate and understand the expected drier conditions caused by global warming, an experimental site was developed at the floodplain area comprising a drainage (hereafter "drained area") and a control site (hereafter "control area"). A drainage ditch with a diameter of about 200 m was constructed in 2004 (Merbold et al., 2009) to promote a persistently lowered water 105 table and test its effects on water transfer and on the carbon cycle regarding future changes of polygonal landscape properties (Liljedahl et al., 2016). The two sites are located in the immediate vicinity, but the control area remains unaffected by the drainage manipulation. Previous investigations on site using eddy covariance (Kittler et al., 2016) and soil chambers (Kwon et al., 2016) have shown differences in carbon dioxide ($CO_2$) and methane ($CH_4$) fluxes between the drained and the control area. The construction of a drainage ring led to drier conditions and a water table decrease of up to 30 cm one year after 110 drainage construction (Merbold et al., 2009) and created shifts in radiation budgets, vegetation patterns, soil thermal regime and snow cover . In this study, we compared hydrological conditions within the floodplain section affected by the drainage with those of the control area.





The hydrological year of the region is characterized by the formation of a snow cover after the growing season. This is followed by an annual snow melt and ice-break up phase in spring. A subsequent spring freshet leads to higher discharge into rivers and

transport of solutes (Fig. A1). Depending on the timing and dynamics of this process, the site usually experiences flooding at the beginning of the growing season, when water levels can rise to more than 50 cm above the surface level (Goeckede et al., 2019). The inundated site is then only accessible by boat. The flood lasts typically for some weeks and recedes by late May to early June. The highest groundwater levels (water above the permafrost, also suprapermafrost) occur typically between May and June, followed by a slow decrease until re-freezing of soils in autumn (September–October). During lowest water levels

(particularly in July and August), precipitation and thawing ice stored in the active layer are the main sources of river water at the floodplain (Guo et al., 2015). Due to active sedimentation during the spring flood, characteristic periglacial formations such as frost polygons are less pronounced within the floodplain.

The topsoil layer is about 15–26 cm depth (Kwon et al., 2016 and soil property data from field work 2018) and consists mostly of organic peat (23 ± 3 cm) formed of remains of roots and other organic material. The underlying silty-clayey mineral layer

originated from river and flood water transport.

The vegetation at the site is categorized as wet tussock tundra (Corradi et al., 2005; Goeckede et al., 2017). The vegetation cover is dominated by cotton grasses (*Eriophorum angustifolium*), tussock-forming sedges (*Carex appendiculata* and *Carex lugens,* e.g., Merbold et al., 2009; Kwon et al., 2016) and shrubs (e.g., *Salix* species and *Betula exilis*, see also Fig. A2). Fractional coverage of these groups roughly follows the soil standing water status, with predominantly wet sites dominated by

cotton grasses, and drier sites dominated by shrubs (Kwon et al., 2016).

Within the measurement period from 12 June 2017 to 22 September 2017 (i.e., 103 days) (Fig. 2), the mean air temperature at the study site was 9.2 ± 5.8 °C (min. temperature: -6.1 °C, max. temperature: 26.9 °C) and cumulative precipitation during this period was 98.4 mm, which represented ca. 67 % of the total annual precipitation in 2017.

## 2.2 Field sampling and laboratory analysis

Air temperature and precipitation data were obtained from sensors installed at the drained area (tipping bucket rain gauge (Thies Clima, Germany) and a KPK 1/6-ME-H38 (Mela) for air temperature). For more details on instrumentation, refer to (Kittler et al., 2016).

We analyzed three parameters to compare water transport mechanisms at the study site: water levels (WL), water stable isotope signals, thaw depths and soil properties in the course of repeated measurement campaigns between 2016 and 2019.

### 2.2.1 Water table depths

We monitored suprapermafrost groundwater and surface water levels within a distributed network of 32 sampling sites (Fig. 1). The sampling sites were placed at representative locations within the drained and control area. In total, 19 piezometer locations were installed within the drained section (sites DI-1 to DI-10 at the drainage inside area; sites DO-1 to DO-9 at the outside area) and 10 locations at the control area (sites C-1 to C-10). Moreover, surface water levels were measured at three





locations within the drainage ring (sites SW-1 to SW-2). The sites were categorized into four main groups: drainage inside area (D-in), drainage outside area (D-out), control area (Ctrl), surface water of the drainage ditch (SW) to indicate the predominant hydrological setting (Fig. 1). An earlier deployment of piezometers was not possible before mid-June due to flooding at the site. Each piezometer consisted of a perforated PVC pipe of 2 m length and a diameter of 110 mm that was installed in the ground and anchored in the permafrost. The water level measurements are based on a downward-looking

ultrasonic distance sensor (MaxBotix MB7380 HRXL-MaxSonar-WRF) installed on top of the pipe, integrated into a custom-built unit that handles data acquisition and power supply. The measurement range of the sensor is 0.3 to 5 m (MaxBotix Inc., 2023a). The batteries were recharged by solar panels. Continuous hourly measurements of the distance to the water table were recorded on a memory card and read out manually at regular intervals throughout the observation period. The data were further aggregated to daily mean values.

We used ultrasonic distance measurements to detect water tables based on the distance between the sensor and the water surface. This method allowed for obtaining water table information even in conditions when the groundwater column was too shallow to measure piezometric heads based on submerged sensors. Such conditions occurred temporarily at dry sites with low active layer depths and low suprapermafrost water bodies, and during periods with no precipitation. In this study, the custom-built devices showed very good results when water levels were close to the surface, but signals were increasingly disturbed

when water levels decreased (July). Measurement errors were mainly linked to scattering of the signal due to distractions in the pipe e.g., water droplets (MaxBotix Inc., 2023b). For some wells, when the water table depth increases, perforation of the pipe itself can result in distracted signals. Other factors influencing the quality of the signal were: obstacles in the pipe, housing with sensor not properly set on top of the pipe, high wind speeds that dislocated the pipe, disturbance due to pipe access (data read out and water sampling) and high temperature changes. During the field campaign, we regularly checked on the data

quality, compared the data with manual measurements and cleaned pipes to minimize measurement errors. However, these disturbances created outliers (Tab. A2) that were filtered semi-automatically according to section-wise minimum values (software R studio, R Core Team, 2023).

   We established a wetness indicator (WI in m) to indicate the relative wetness degree on each site (dry vs. wet conditions) on the basis of the water table depth data. We used data covering the measurement period in 2017 (June to September) and of all

measurement locations to conduct a cluster analysis with two classes (software R studio, package *stats*, function *kmeans*). These two classes represented the relatively dry and wet piezometer locations and the threshold value could be determined with: WI > –0.138 m for wet conditions, and WI <= –0.138 m for dry conditions (Tab. A1). The wetness indicator is given in m in relation to the ground level (GL), where positive values represent water tables above and negative values below GL.





**Figure 1: Map of the distributed network of water level depth monitoring locations at the Ambolikha observation site. DI-wells show piezometer locations at the drainage inside (D-in) and DO-wells at the outside (D-out) area, C-wells at the control area (Ctrl). Water levels were measured and sampling conducted at three surface water (SW) locations within the drainage ditch (marked in yellow). Black points show automatic water level measurement locations and white points indicate water sampling at several of these locations. Red diamonds indicate soil sampling locations for porosity analysis. The background map is based on WorldView-2 2011. Data of the overview map is based on GADM (2023).**



In order to determine the absolute water level above sea level, we obtained the elevation of each of the monitoring locations across the Ambolikha site based on a 2018 drone survey that produced high-resolution digital elevation maps of the surface and top of the piezometer pipes with a precision of ± 6 cm. The level of the ultrasonic sensor within the pipe and the soil surface adjacent to it were measured manually to account for different exposure heights of the piezometers. Our piezometers protrude ca. 78 cm (± 10 cm) from the soil. Based on this information, we were able to calculate the network-wide spatio-temporal variation of groundwater heads above sea level (groundwater level, h), as well as the depth to water table from the surface (relative water table depth) for groundwater and surface water.

In order to visualize spatial water level trends on the basis of the piezometric data, we first applied cubic spline interpolation (QGIS.org, 2022) on all surface and groundwater levels for four dates throughout the measurement period (Fig. 5). Mid-monthly dates were used to have an overview on water levels throughout the growing season. The main water flow directions were illustrated using the tool *gradient vectors from surface* in QGIS (QGIS.org, 2022).

Suprapermafrost water flow ($Q_w$ in $m^3\ s^{-1}$) between piezometers was calculated with Darcy's law and given in $L\ d^{-1}$:

$$Q_w = K \times A \times \frac{\Delta h}{\Delta x} \tag{1}$$

where K is the saturated hydraulic conductivity ($m\ s^{-1}$), A the cross-sectional flow area ($m^2$) based on groundwater level above the permafrost, $\Delta h$ the water level difference between piezometers (m), and $\Delta x$ the lateral distance between piezometers (m).

### 2.2.2 Soil properties

Saturated hydraulic conductivities (K) were determined based on slug-injection tests, which were conducted on 29 July and 30 July 2016. K was calculated according to the description of (Bouwer and Rice, 1976) resulting in two different values: 2.5 x $10^{-5}\ m\ s^{-1}$ for the organic and 7.4 x $10^{-7}\ m\ s^{-1}$ for the mineral soil layer. Depending on the amount of water located in the organic or mineral layer, the effective hydraulic conductivity (weighted mean) was calculated from the contribution of each layer to the active cross section and applied in the Darcy flow calculation.

The extent of the organic layer was measured on 08 July 2018 and 17 July 2018. This was done by drilling six small holes using an auger within the drained and control area. The transition between the organic and mineral soil layers were visually detected and measured. At these locations, samples for soil porosity measurements were collected in core cutters of 100 cm³ volume. The soil samples were transferred to a laboratory on-site and were weighed twice: first after two days under water saturated conditions. Subsequently, the soil samples were dried at 105 °C for 24 h and weighed again. The porosity was calculated using the relationship between water volume and soil weight under consideration of the core cutter volume:

$$V_p = V_t - V_s \tag{2}$$

where the pore volume is $V_p$ ($g\ cm^{-3}$), $V_t$ is the total volume ($g\ cm^{-3}$) and $V_s$ the solid volume ($g\ cm^{-3}$) of the respective soil sample.

$$P_t = \frac{V_p}{V_t} \times 100 \tag{3}$$

where $P_t$ is the soil porosity (%) that was calculated from the ratio between $V_p$ and $V_t$.



### 2.2.3 Thaw depths

Thaw depths were measured by inserting a metal rod into the soil until the ice layer was reached (no further movement possible)
and the distance from the soil surface was noted. Thaw depths were measured in the majority of the groundwater monitoring
locations as well as distributed over the study site and repeatedly during the course of the growing season. For most of the
groundwater monitoring locations up to eight thaw depth measurements were conducted between 17 June 2017 and 04
September 2017.

### 2.2.4 Water stable isotopes

For stable water isotope analysis, water samples were collected from precipitation, surface water, suprapermafrost groundwater
and the upper permafrost ice layer. Rain water was collected after rain events during the field sampling period. In total, three
surface water measurement locations and 16 groundwater piezometer sites were sampled for this analysis, five of which are
located within the control area and five within the drained outside and six within the drained inside area (Fig. 1). 14 samplings
were conducted in the growing season between 25 June and 05 September 2017. During that time, precipitation was sampled
eight times subsequent to precipitation events. The permafrost ice was sampled on two dates in 2018: 03 and 06 July. Generally,
the temporal resolution in the middle of the growing season was good, but lower for mid-August and mid to end of September.
The inclusion of additional data from July to August 2016, July to October 2018 and May to July 2019 covered early and late
growing season isotopic signatures and provided sufficient data for monthly means (Fig. 7). This additional data resulted from
own measurements on-site in 2016 and 2018, suprapermafrost groundwater measurements on-site in 2019 and precipitation
measurements in Chersky from 2018 and 2019. During all 14 water samplings, electrical conductivity among other parameters
were measured with a YSI Professional Plus multiparameter instrument combined with the respective parameter sensors
(Xylem Inc., USA).

Water samples were collected, filtered (0.7 μm GF/F Whatman®) and transferred into 1 ml glass vials without headspace and
kept at 8 °C prior to analysis. Permafrost ice water was sampled by drilling boreholes to the uppermost part of the active layer
and melted prior to filtering and measurement. All water samples were analyzed for hydrogen ($\delta$D) and 18-oxygen stable
isotope composition ($\delta^{18}$O), with a Delta+ XL isotope ratio mass spectrometer (Finnigan MAT) at the BGC-IsoLab of the Max
Planck Institute for Biogeochemistry in Jena, Germany, using a TC/EA (thermal conversion elemental analyzer) technique.
Isotopic compositions relative to the Vienna Standard Mean Ocean Water (VSMOW) and to the Standard Light Antarctic
Precipitation (SLAP) are expressed as $\delta$-values in ‰ (Coplen, 1994). The BGC-IsoLab further used three internal standards
(www-j1, BGP-j1 and RWB-J1) and analytical uncertainties are about <1 ‰ for $\delta$D and 0.1 ‰ for $\delta^{18}$O (Gehre et al., 2004).
We set up an end member mixing analysis (EMMA) to detect if the isotopic signal was derived from precipitation or snow
melt (in the beginning of the season) or permafrost melt (in the end of the season) signal. Since permafrost and snow melt
showed a similar heavy direction in isotopic compositions, we can distinguish them only by early and late season.

$$proportion\ of\ source\ 1 = \frac{(sample - source\ 2)}{(source\ 1 - source\ 2)} \times 100 \qquad (4)$$



where source 1 (end member 1) represents the stable water precipitation signal (n = 7, $\delta^{18}O$ = –15.3 ± 0.7 ‰, $\delta D$ = –118.2 ± 4.4 ‰) and source 2 (end member 2) the heavier signal of permafrost ice (n = 2, $\delta^{18}O$ = –22.8 ± 0.2 ‰, $\delta D$ = –180.8 ± 2.6 ‰). The proportion of source 1 is given in %.

Deuterium excess (D-excess) was calculated in order to assess kinetic fractionation processes (Dansgaard, 1964). These processes gave hints about evaporation and condensation processes of the samples. Samples with a deuterium excess <10 ‰

represented an evaporative signal (Dansgaard, 1964).

$$D - excess = \delta D - 8 \times \delta^{18}O \qquad (5)$$

where $\delta D$ and $\delta^{18}O$ represent the stable water isotopic values per site and sampling time.

## 3 Results

### 3.1 Water table depth and water levels

Water tables for each of the four groups of water types are significantly different (p-values < $2.2e^{-16}$, software R studio, package *stats*, function *t.test*). For suprapermafrost groundwaters, water tables were highest for control site wells, followed by drainage outside areas and drainage inside areas (Fig. 2). After the recession of the early summer flood, surface and groundwater levels receded across all sites. While most of the locations still show inundation in June, these relatively wet conditions are followed by a decrease in water levels until mid to end of July, most pronounced within the areas affected by the drainage. Accordingly,

around the peak of the growing season in mid-July the largest fractions of the drainage area, but also some locations within the control area (C-4 to C-6), have dry topsoil. After reaching lowest water levels around early August, conditions remain relatively stable for the rest of the measurement period, followed by a minor to moderate increase linked to more frequent precipitation events and lower temperatures in September (see also Fig. 3).







**Figure 2: Daily mean water table depths (water tables in relation to ground level 0 m (GL), horizontal black line) in 2017 for all measurement locations. The individual panels show time series for each of the four hydrological sections (Tab. A1): (a) DI-wells, (b) C-wells, (c) DO-wells, (d) SW-wells. Values above the ground level indicate periods of waterlogged conditions. For the surface water locations, the ground level refers to the water–sediment interface.**

Dry and wet locations reacted differently to precipitation with different temporal fluctuations of piezometric heads during the course of the growing season. In general, all precipitation events were associated with an almost immediate increase in groundwater levels across sites. For example, water levels at a predominantly wet site (C-01) for the entire observation period mostly remained within a range of ± 6 cm, and daily fluctuations rarely exceeded 0.3 cm (Fig. 3). Strong precipitation events slowed down or even reversed the general drying trend over time, but observed increases in water levels usually happened slowly over the course of several days. In contrast, at the drained site (for example DI-08, Fig. 3) we observed water tables as low as 30 cm below soil surface, and steep decline rates partly exceeding 1 cm per day. There, strong precipitation events were followed by an instantaneous rise in water levels exceeding 5 cm several times during the observation period. Overall, the measured groundwater level range was about three times higher at drained sites compared to control sites.



**Figure 3: Temperature, precipitation and representative water tables over the course of the measurement period in 2017.**
**Precipitation events are shown as vertical blue bars. Yellow shadings show representative dry periods (no precipitation for more**
**than one week) and blue shadings precipitation events. (a) time series of the air temperature during the measurement period. The**
**grey line shows hourly data, the black line the moving average (width of rolling window = 100 data points, *zoo* package R Core**
**Team, 2023) temperature values. (b) time series of precipitation events on two exemplary locations of ground water levels. The red**
**line gives water levels at a selected drained site (DI-08), while the grey line indicates conditions at a control site (C-01). In both cases,**
**water levels are given as depth from ground level (GL, horizontal black line) to water table.**

We observed two representative dry periods in 2017, where precipitation was absent for more than one week: in July (from 10

July 2017 at 19:00:00 LT to 19 July 2017 at 12:00:00 LT) and in August (05 August 2017 at 07:00:00 LT to 16 August 2017

at 21:00:00 LT, all are local time) (Fig. 3). Daily mean temperature during these dry periods rose up to 18 °C in July, but only





11 °C in August. The water level decrease rates differed distinctively between the wells. For instance, the water level at C-01

290 decreased by 3.2 cm in July and 2.5 cm in August. In contrast, water levels at the drained site DI-08 decreased by 10.3 cm and

9.5 cm, respectively. At the same place a strong precipitation event on 17 August 2017 at 01:00:00 LT (3.5 mm h$^{-1}$) induced

an increase of 8.0 cm in water level within 3 h after the start of the event. At C-01, the strong precipitation event only resulted

in 1 cm water level increase with no or very limited lag time. Another strong precipitation period started on 08 September

2017 at 14:00:00 LT. During this event, C-01 and DI-08 showed similar increases in water levels (C-01: 39 mm and DI-08: 5

cm, Fig. 3), while in contrast to the August event the peak was delayed at the control site C-01, not at the drained site DI-08.

Overall, daily groundwater levels were highest in the morning (ca. 07:00:00–11:00:00 LT) and lowest in the evening (ca.

19:00:00–23:00:00 LT), indicating evapotranspirative losses during the day. These fluctuations can be observed for all sites,

but were more pronounced at drainage inside sites (data not shown).

**3.2 Water flow patterns**

The highest piezometric levels at the control area were located in the north, while the lowest water levels were generally

observed within the center of this section, indicating a potential lateral subsurface outlet (Fig. 4). Across the site, water levels

tended to decline until August and then slightly recovered. Linked to this, the flow patterns showed a pronounced variability

over the course of the growing season. In general, water from the wetter areas in the north and the south flowed towards a

convergence zone in the center. The position of the central convergence zone shifted with time from the southern part (C-07,

C-08) towards the north until mid-August, and then back again south until the end of the observation period. Accordingly,

during high water levels the main outlet for water from the study area is located close to C-08, while with lower water levels

water flows rather towards C-06, and is drained down the surrounding floodplain from there.







**Figure 4: Interpolated piezometric levels and flow directions for four selected dates (15 June, 15 July, 15 August and 15 September)**
**across the growing season 2017 at the control and drained area. Water level measurements in suprapermafrost water and surface**
**water in the drainage ditch are shown with black dots. Flow directions are marked as arrows and contour lines of absolute water**
**levels in grey lines. Interpolated water levels are indicated by color code (low water levels in red, high water levels in blue), note the**
**difference in color scale for the control and drained area due to general differences in absolute heights. The background map is**
**based on WorldView-2 2011.**





Within the drainage area, the spatial distribution of high and low piezometric levels retained a similar pattern throughout the measurement period: we observed highest groundwater levels in the north-western area, while the lowest ones occurring close to the outlet of the drainage ring in the south-eastern part (Fig. 4). Locations inside the drainage ring showed the strongest temporal fluctuations, especially in the south-eastern part where water levels were lowest.

Based on hydraulic groundwater gradients between sampling sites, we determined the expected main flow direction within the

drainage ring to be oriented from northwest to southeast, e.g., groundwater flow towards the outlet drainage channel (especially towards SW-3, Fig. 4) and the Ambolikha River. The convergence zone was found in the south-east at the drainage ring area, around the connection to the outlet channel. During the drier months July and August, lower water levels generally intensified the flow paths; however, the overall flow patterns within the drainage area remained stable over the course of the growing season, even though absolute water levels as well as their gradients within this treatment area changed over time.

The calculation of Darcy flow showed a relatively consistent pattern for drainage inside areas. Darcy flows varied between reversed flow (-0.04 L d$^{-1}$) between piezometer locations and 0.88 L d$^{-1}$. For many piezometer sets, high flow velocities were calculated for June, the lowest in July and highest in September. Flow rates remained persistently high in some areas, while particularly for the drainage inside area lowest flow gradients were found during mid-summer. Flow velocities within the control areas showed different patterns depending on the location of the piezometers (Table 1), including both permanent

decline over the summer, as well as peak flow or low flow in mid-July. In general, highest water flows were calculated for drainage outside and drainage inside sites followed by the control area.

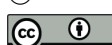



**Table 1: Suprapermafrost Darcy flow velocities [L d⁻¹] calculated for three time steps in 2017.**

| Area | Darcy flow direction | Darcy flow velocity [L d⁻¹] | | |
|---|---|---|---|---|
| | | **17 June 2017** | **17 July 2017** | **04 September 2017** |
| **D-in** | DI-06 to DI-09 | 0.88 | 0.36 | 0.43 |
| | DI-01 to DI-02 | 0.18 | 0.25 | 0.29 |
| | DI-03 to DI-04 | 0.29 | 0.12 | 0.38 |
| | DI-08 to DI-10 | 0.27 | 0.03 | 0.21 |
| | DI-06 to DI-02 | 0.30 | 0.14 | 0.13 |
| | DI-01 to DI-04 | 0.15 | 0.03 | 0.22 |
| | DI-07 to DI-08 | 0.16 | 0.02 | 0.13 |
| | DI-08 to DI-09 | 0.10 | 0.02 | 0.12 |
| | DI-04 to DI-02 | 0.16 | -0.02 | 0.15 |
| **D-out** | DO-01 to DO-02 | 0.53 | 0.32 | 0.26 |
| | DO-03 to DO-04 | 0.36 | 0.35 | 0.30 |
| | DO-05 to DO-06 | 0.05 | 0.04 | -0.02 |
| | DO-07 to DO-08 | 0.01 | 0.02 | 0.03 |
| **Ctrl** | C-01 to C-03 | 0.28 | 0.19 | 0.15 |
| | C-01 to C-04 | 0.11 | 0.11 | 0.03 |
| | C-10 to C-08 | 0.39 | 0.02 | 0.05 |
| | C-02 to C-04 | 0.11 | 0.09 | 0.03 |
| | C-09 to C-10 | 0.06 | 0.04 | 0.03 |
| | C-04 to C-06 | 0.05 | 0.01 | 0.04 |
| | C-08 to C-06 | 0.001 | 0.09 | -0.04 |
| | C-07 to C-05 | 0.01 | 0.04 | -0.04 |
| | C-08 to C-07 | 0.01 | 0.05 | 0.01 |
| | C-05 to C-06 | -0.01 | 0.001 | 0.01 |
| | C-04 to C-05 | 0.0000000001 | 0.0004 | 0.01 |

**3.3 Soil water saturation**

Thaw depths (Fig. 5) showed an initial steep decline at dry sites, which then stabilized in late summer (49.0 ± 12.4 cm on 04 September 2017). In contrast, wet sites were characterized with a lower initial decline in thaw depths, but continuous deepening of thaw levels ultimately led to generally deeper thaw in September (62.7 ± 8.0 cm).





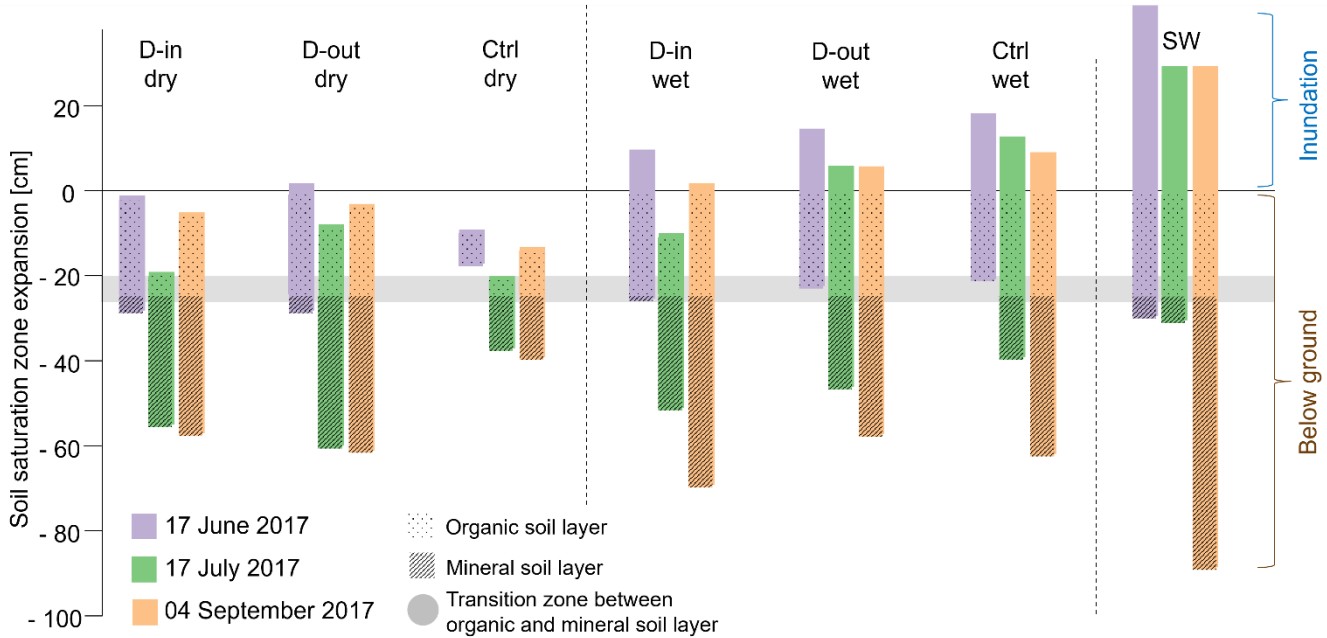

**Figure 5: Water saturation zones for three measurement times (17 June, 17 July and 04 September 2017), for different water types (D-in, D-out, Ctrl, SW) and differentiated between wet and dry areas. The grey area represents the transition zone from upper organic to lower mineral soil layer (D-in dry n = 6, D-out dry n = 1, Ctrl dry n = 2, D-in wet n = 4, D-out wet n = 8, Ctrl wet n = 8, SW = 3; see Tab. A1). Water saturation zones were calculated using water table, as the top hydraulic boundary, and ice table (thaw depth data) as the bottom boundary.**

Water levels declined continuously only at very wet sites. For all other areas, there was a minimum water level in mid-summer, followed by an increase. This was emphasized in the drained inside area (dry and wet), where the initial drop in water levels was steepest. This slightly recovered in September, while most of the control area water levels continued declining (Fig. 5). Mid-June, mid-July and early September data revealed differences in the overall location as well as the dimension of the saturation zone (Fig. 5). The extension of the saturation zone is generally lowest in June. Low thaw depth tables in June allowed the saturation zone to mainly extend into the organic soil layer, even though water levels are highest. In contrast to dry sites (Tab. A1), wet areas showed highest inundation in June. The size of the saturation zone in July increased slightly for wet areas and substantially for dry areas and was shifted downwards into the mineral soil layer. The largest extent of the saturation zone can be found in September, where the increase was mainly resulted from water levels at the dry sites and from thaw depths at the wet sites. Generally, more extensive total saturation zones were found at wet sites (ca. 10 cm larger at the drained area and ca. 37 cm larger at control sites) in contrast to dry sites. Surface waters showed highest water levels and a large increase in thaw depths in the late season.

The transition zone (20–26 cm below ground) separated the upper organic from the lower mineral soil layer. The mean porosity of the organic layer was high, at 79.0 ± 3.6 %, and low for the mineral soil layer 24.1 ± 2.0 %.





### 3.4 Stable water isotopes

The water isotope data showed two main patterns: a) temporal differences, indicating that the mean June samples were most depleted in δD and $\delta^{18}$O; and b) spatial differences between water types. That latter highlighted that, in the period July through September, surface waters were less depleted, followed by drainage outside areas, control area and drained inside areas, even though the differences between suprapermafrost groundwaters was relatively low (Fig. 6). Apart from the June data and in most areas, the strongest depletion was observed in August, while in the drainage inside this peak was already reached in July.

The lowest range was found for drainage inside locations. The largest shift was found for surface water isotopes between June and July.

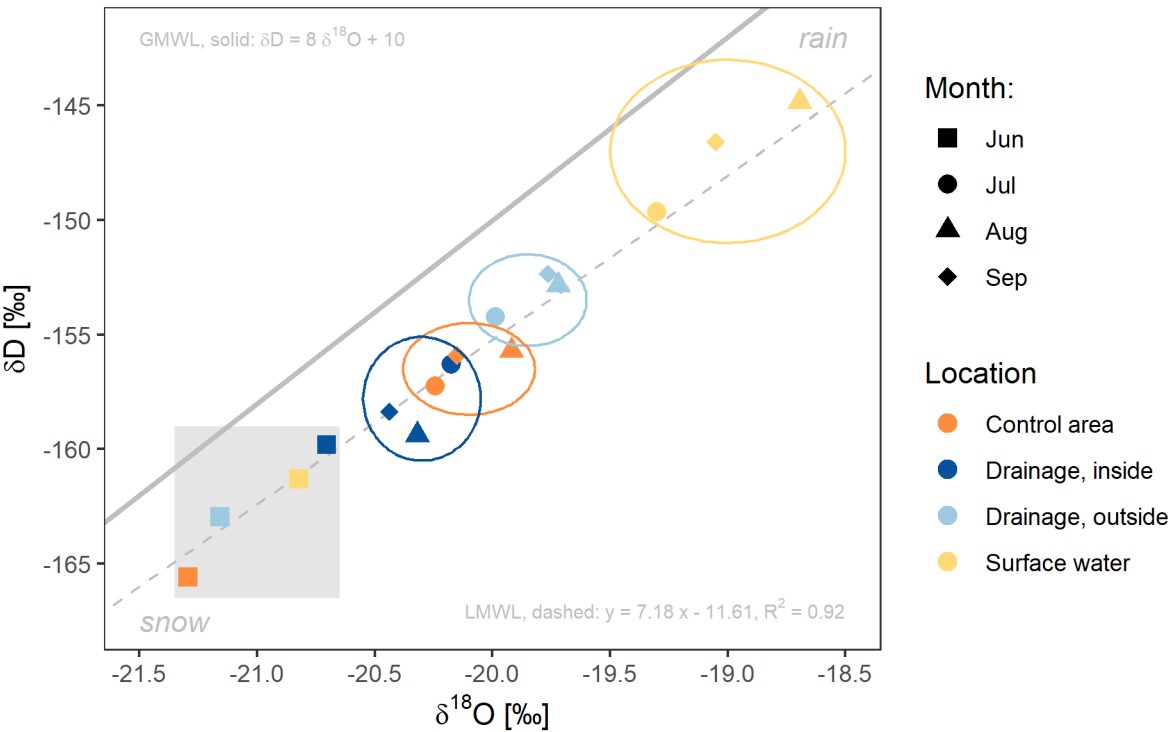

**Figure 6: Mean stable surface water and suprapermafrost groundwater isotopes measured from 2016–2019. Color code shows the different water type locations, shapes represent the respective month data. Circles in the respective colors visually summarize the**
**months (Jul–Sep). The grey shaded rectangle shows all June samples. The solid line represents the Global Meteoric Water Line (GMWL: δD = 8 x $\delta^{18}$O + 10) and the dashed line the Local Meteoric Water Line (LMWL: y = 7.18 x -11.61; $R^2$ = 0.92, own measurements).**

The end member values used for the calculation of the local meteoric water line (LMWL) was for permafrost ice ($\delta^{18}$O = − 22.8 ± 0.2 ‰, δD = −180.8 ± 2.6 ‰) and for rain ($\delta^{18}$O = −15.3 ± 0.7 ‰, δD = −118.2 ± 4.4 ‰) and were in close agreement
with the values reported by Welp et al., (2005). The average composition of the sampled water in the system was 42 ± 8 % of precipitation and 58 ± 9 % of snow/permafrost melt water (Fig. 7). Surface waters generally indicated a decrease in snow melt water signal, and simultaneously an increase in rain water signal, with time. A similar trend with continuously rising





contributions from precipitation water over time was found at the wet locations DO-01, DO-07, DO-03 at the drainage outside area and C-01, C-03, C-10 within the control area. In contrast, most of the drainage inside sites, and also dry to intermediate

sites in other areas (such as DO-09, DO-05 and C-06), initially showed a decrease in snow melt water signal, followed by a substantial increase in permafrost water towards the end of the sampling period.

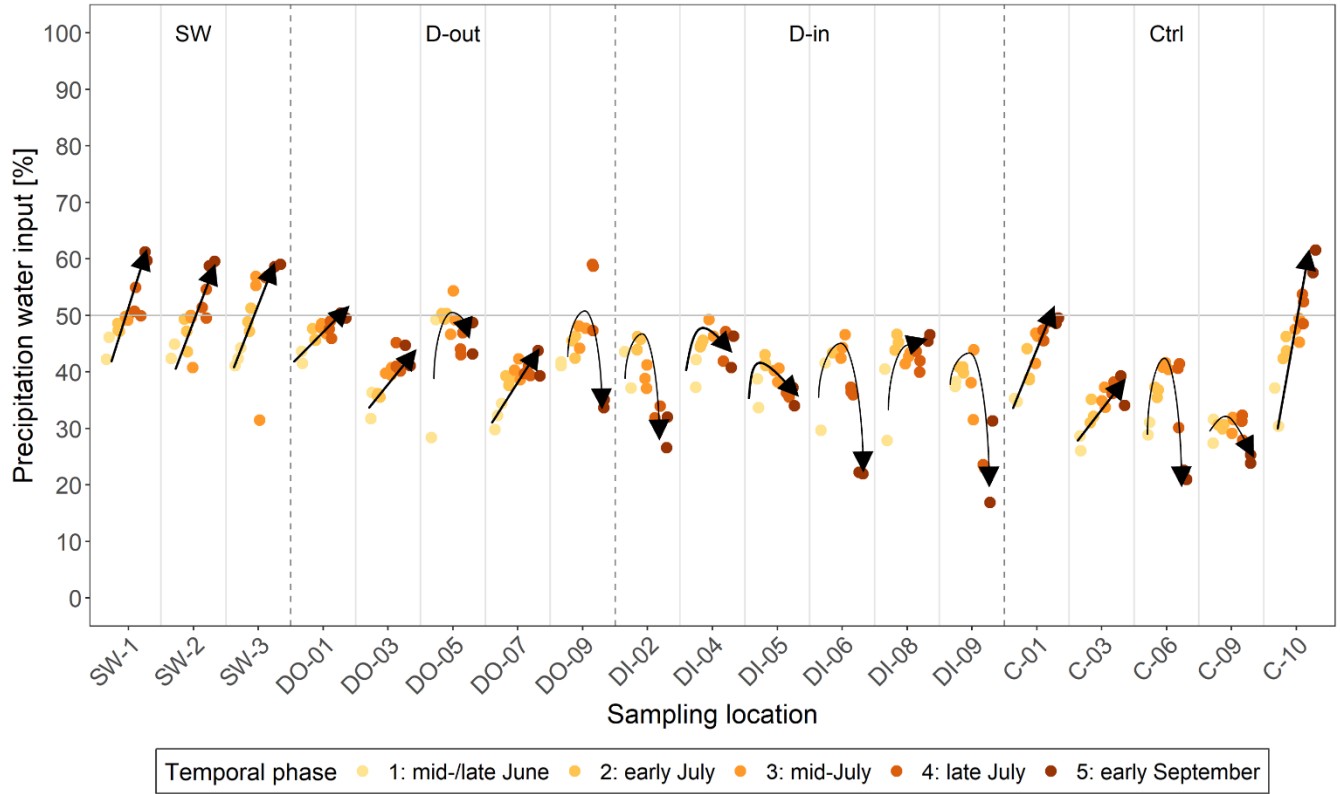

**Figure 7: Percentage of precipitation input (end member mixing analysis) for δD at the samples during the measurement period in 2017. The lower the percentage, the higher is the influence of the stable water isotope signal from snow melt water in early season**
**(spring freshet) and permafrost melt water in late season measurements.**

## 4 Discussion

This study provides detailed information about observed shifts in suprapermafrost groundwater conditions comparing a control and a drained area in the Kolyma floodplain near Chersky. The artificial drainage at the study site results in changes in hydrological conditions in contrast to a near control area, affecting e.g., water table depths, water flow velocity and transport

patterns. These shifts in surface and soil water regimes have secondary disturbance impacts on other ecosystem characteristics, including thaw depths and vegetation communities. The resulting shifts in residence times of water and soil water saturation status might constitute an important driving factor for the lateral mobilization of carbon in this site.



## 4.1 Degree of lateral connectivity impacts water level fluctuation

The study site is located in a floodplain adjacent to the Kolyma River, and characterized by shallow topography. As a
consequence, the studied area was dominated by inundated and wet soil conditions in its natural, undisturbed state (control
area). The water table depths at the control area were characterized by high water levels with some wells that were inundated.
Also, most water levels dropped belowground later in the season and inundation occurred only sporadically (linked to rainfall
events) and remained close to the ground. This can be associated with the general wetness status (Tab. A1) and the lateral
connection throughout the areas (Lamontagne-Hallé et al., 2018). Still, a clear difference could be observed at the dry locations
C-04 and C-05 of the control area where water table depths were lower due to higher elevated topography. This also led to a
development of shrubby vegetation at these sites (Fig. A2). The site C-06 represented an intermediate state between dry and
wet areas following the more distinct water level trend as C-04 and C-05, but with a higher water table depth. The drained
area, however, was affected by significantly lower water table depths, and therefore larger parts of this domain were
characterized by drier soil conditions in the measurement period. Water table depths decreased distinctively from mid to end
of July, where precipitation was absent and evaporation rates high due to higher air temperature. Furthermore, daily, seasonal
and precipitation-based fluctuations in water levels were more pronounced. Lowest water levels occurred in July, where the
uppermost soil layer became drier. Most of the drainage outside area showed wet conditions with smoothed water level trends,
similar to the control sites. Surface waters within the drainage ditch never dried out and were well connected and all piezometric
heads showed the same trends.
The contrasting water levels between the area inside the drainage and the control area led to different responses to precipitation
events. As a result of the generally high water levels, water table trends at wet sites were smooth, and the influence of short-
term precipitation was dampened in comparison to dry sites. After four precipitation events (Tab. A3), the overall median
water level increase was 0.049 m for drainage inside areas, 0.01 m for control areas and 0.018 m and 0.022 m for drainage
outside and surface waters, respectively, highlighting the influence of precipitation events on drained inside areas.
Accumulation of precipitation events had a long-term influence (Fig. 3) with increases in water levels at all sites, but this signal
was delayed. The increased lag time was a result of the laterally connected wet regions at the control area. Because of lower
water levels above the frozen ground layer in the drained area, water levels were not as laterally connected as in the control
area, and the increase in water levels was more than three times greater for short-term precipitation events during driest
temporal periods. The capacity of rainwater to infiltrate into dry soils was faster than the lateral discharge towards the drainage
ditch (Frampton and Destouni, 2015). During long-term precipitation events this effect was not that pronounced due to general
higher water levels, a larger saturated zone and an increased lateral connectivity. The soil water capacity was reached at wet
sites with water saturated soils or inundated areas, therefore the rain water flowed in the upper part of the organic layer and
surficially. Water was redistributed over the area and slowly moved towards small channels and topographically lower areas
discharging into the Ambolikha River. During periods without rainfall, higher evaporation rates combined with an increase in
air temperatures, led to lower water levels. Limited potential groundwater recharge but water flow following hydraulic





gradients (Walvoord and Kurylyk, 2016), were the main processes affecting the water table depths during this time. Generally, precipitation input is more dominant than differences in temperature, even though temperature often dropped when rain fell (Fig. 3).



**Figure 8: Schematic water levels and thaw depths for three measurement times (17 June, 17 July, 04 September 2017, data in m). From left to right the schematic shows the drainage inside (D-in), surface water (SW), drainage outside (D-out) and the control (Ctrl) area.**



Water levels and flow patterns follow a characteristic area-specific structure (Fig. 2–4, 8), where hydraulic gradients as well as soil saturation seemed to be the main drivers (O'Connor et al., 2019), although precipitation plays a short-term important role, except from consistent precipitation periods (e.g., September 2017). The more the soil area was saturated, particularly in the organic layer, a greater potential for lateral connectivity throughout the area was facilitated. During the spring freshet in June, water flow was limited to the organic soil layer due to shallow thaw depths (Fig. 8) as permafrost represents an impermeable barrier (Grannas et al., 2013; Vonk et al., 2015). Surface water flow, inundation at wet sites and groundwater discharge within the organic layer played a major role during this period (Woo and Young, 2006). At dry sites, transport was limited to the organic soil layer without inundation, where water flow is less vertically and laterally connected (Koch et al., 2013). Wet areas in July also showed much better connectivity compared to dry sites, even though the standing water level decreased during this period. Still, the abundant soil water in the organic and mineral layers allowed for efficient lateral transport processes.

Between early to mid-July, the lowest water levels were measured at all piezometer sites. With the spring freshet having fully receded at this time, individual flow patterns appeared, and lateral connectivity of water decreased overall. This was strongly enhanced due to dry conditions with warm temperatures in July (Fig. 3). A vertical soil water exchange at the drained area was inhibited due to very low water levels, which were mainly located in the less permeable mineral layer. During this time water flow was lower and mainly located at the transition zone between organic and mineral soil layer. Residence times relatively increased and water redistribution slowed down (Table 1), since water levels were mainly located within the less permeable mineral layer or at the transition zone between both soil layers. Infiltrated precipitation water could be accumulated at this transition zone and quickly discharged laterally before fully percolating into the mineral layer (Koch et al., 2014; Walvoord and Kurylyk, 2016; Wright et al., 2009).

In September, linked to the enhanced precipitation input, the vertical and horizontal water connectivity at drained sites increased, and the input water was redistributed in both the organic and deeper mineral layers (Fig. 8). Groundwater recharge due to percolated precipitation was more prominent during that period, enhanced also by lower air temperatures and therefore lower evaporation (Fig. 3). Also, limited photosynthetic activity in the end of the growing season led to limited water uptake. For control sites, the connectivity remained high both laterally and vertically (complete organic layer, inundated water on top and a stronger input from mineral soil layer), even though inundation generally decreased over the course of the growing season.

## 4.2 Water flow depends on micro-topography and position of the water table in the soil column

Water flow direction and speed were dependent on several parameters. These included e.g., the amount of available water (mainly from inundation and precipitation), topography, and the location of the water table within the soil (Gao et al., 2018). Soils at our study site consisted of an organic peat layer on top with a subjacent silty-clayey mineral layer. The organic layer with high pore volumes and high hydraulic conductivities facilitated water flow, whereas the mineral layer restrained water flow due to low pore volumes and low hydraulic conductivities (Walvoord and Kurylyk, 2016). The lateral redistribution of





water within this site varied depending on the water table depth and thaw depth location, as well as the resulting position of the soil saturation zone.

Microtopographic features led to a formation of local elevations and depressions on both areas, which had a profound impact on the redistribution of water at small scales. Measurement and sampling sites situated at local elevations (e.g., DI-03, DI-01, 470 C-04, C-05) had relatively dry soils, whereas sites within local depressions (e.g., DI-10, DO-02, C-03, C-07) showed wetter soil conditions throughout the study period. Due to microtopographic features, the soil composition was different at local depressions compared to local elevations. Higher elevated sections were characterized with a comparably larger acrotelm layer (O'Connor et al., 2019), which is the uppermost organic layer with actively decomposed material and high permeabilities, but with a thinner organic layer overall. In contrast, in lower elevated sections, the catotelm layer with denser peat formation and 475 a lower permeability was larger than at the higher elevated sites (O'Connor et al., 2019). In this study, we measured the depth of the total organic layer and identified the transition zone between organic and mineral soil layer. Despite the fact that the distinction between acrotelm and catotelm (O'Connor et al., 2019) was not the focus of this study, explaining water flow velocities and hydraulic conductivities by only distinguishing between organic and mineral layer was sufficient. High hydraulic conductivities within the organic soil layer resulted in a potential faster water flow into the nearby drainage ditch (Hinzman et 480 al., 1991; Quinton and Marsh, 1998), especially during the spring freshet, when water levels were mainly located within this layer. Control area locations showed generally slower flow velocities, possibly affected by a thicker catotelm layer. Dry local elevations as well as the drained area might be more influenced by thicker acrotelm layers, and therefore more pronounced flow paths and quicker flow speeds can develop (Table 1).

The main water flow followed the hydraulic gradient from high to low areas correspondingly to topographical features 485 (Walvoord and Kurylyk, 2016). For the drained area, this gradient was intensified by the construction of the drainage channel. At the drained area the water was first directed to the discharge areas and flowed towards the outlet of the drainage ring to finally discharge into the Ambolikha River. Such a lateral surface connection in a channeled flow can be found at degraded polygonal tundra systems (Liljedahl et al., 2016; Serreze et al., 2000), a process which is intended to be reproduced with the installation of the drainage system at the site (Goeckede et al., 2017; Merbold et al., 2009). Inside the drainage ring, the main 490 flow direction followed the gradient from higher to lower elevated areas, with water mainly entering the drainage channel around site DI-09 (and DI-10, SW-3). This site was the closest to the drainage ring outlet and with very low general water table depths, and potentially low water residence times (Koch et al., 2013). This is the main location where most of the suprapermafrost water with its constituents left the inner drainage system and transitioned into surface waters. The autochthonous carbon at the drainage channel is then further transformed (e.g., assimilated by microorganisms, oxidized) and 495 transported within the drainage channel.

The calculated suprapermafrost water flow at drained sites was faster in June and September compared to the control area following hydraulic gradients (Table 1). In July, when water levels decreased to the minimum measured during the season, water flow was limited to the less permeable mineral layer and the water column within the soil was smallest. During that time, discharge into the drainage channel was lower and short-term influenced by precipitation events although the flow direction





500 remained the same. This is consistent with the observation that the convergence of flow within the drainage area around site DI-09 was much more intense during warmer summer months (July and August) (Fig. 4).

The main flow direction at the control area was from the north and towards the Ambolikha River which represented the overall hydraulic gradient, although a dry hummock was identified at the area (C-04, C-05 and C-06). South of this hummock the water flow direction in June and September was sidewards following small belowground flow paths, with slow water flow

505 speeds and therefore higher residence times with respect to drainage sites. During lowest flow in July and August, lateral water export shifted towards the north (Fig. 4), probably due to small impediments on site and roughness in soil texture. This also led to changes in ephemeral small belowground drainage channels (Connon et al., 2014). Seasonal shifts in preferential flow paths could change the input in carbon concentration (e.g., transport of previously accumulated carbon). Most of the wet areas can be associated with local depressions and confluence sites, where groundwater accumulated from the surrounding area and

510 was slowly laterally exported (Connon et al., 2014). Location C-10 was directly affected by the Ambolikha River and discharged towards it throughout the measurement period.

Permanent inundation and very wet soil conditions lead to a saturated organic layer over the growing season with slow water movement and relatively long residence times. Such long residence times were assumed for the control area, where the water flow was generally lower (Table 1). Longer residence times can be associated with larger vertical flow paths due to percolation

515 which is more pronounced when the active layer deepens. This is intensified in contrast to lateral flow and can be evidence for longer residence times in comparison with the drained section (Frampton and Destouni, 2015; Koch et al., 2013). The longer residence times with water moving slower over the area as well as the different wetness statuses could have a direct influence on carbon production (anaerobic vs. aerobic) and export (e.g., direct vertical release from inundated water column, (Dabrowski et al., 2020; Wang et al., 2022)). With drainage, large parts of the organic layer become dry in summer and initial water from

520 snow melt as well as from precipitation discharge with comparably short residence times.

Porosity within the mineral layer was more than three times lower compared to the organic layer (Tab. A4). Potential water flow was therefore enhanced in the organic layer and highest suprapermafrost groundwater discharge dominantly within that zone (Connon et al., 2014; Walvoord et al., 2012). Most sites dried completely from June to July due to the decreased freshet water, increased air temperatures and lack of rainfall. Although the thaw depth increased, the organic layer became very dry

525 and the remaining pore water in the mineral layer was at its minimum at dry and drainage sites (Fig. 5). The drained inside areas experienced a gain in water in both soil layers between July and September with the potential to enable carbon export through the system. At control wet sites a gain of water within the mineral zone could be detected, but a water loss in surficial water also took place at the same time.

The abundances of the stable water isotopes measured in this study served as indicators of the seasonal composition and

530 transition of the surface water and groundwater influenced by evaporation, presence of snow and precipitation events, and helped identify pathways for lateral water transport in both study sites. The temporal trend at drained sites showed a clear shift from a snow-melt dominated signal at the beginning of the study (i.e., more depleted $\delta^{18}O$ and $\delta D$ values) that decreased over





time and was replaced by permafrost thaw signal at the end of the measurement period. Control sites were more influenced by the precipitation signal and accumulated water flow throughout the area.

In July, the stable isotope composition indicated an increase in the relative contribution of the rain water signal (Ala-aho et al., 2018). Towards the end of the measurements (August and September), the patterns between control and drained areas became distinctive. Sites that were well connected vertically and laterally, with high to inundated water levels, had a dominant presence of rain water. In these sites, the stable isotopic compositions were less depleted and more in contact with the surface and therefore more prone to evaporation (Welp et al., 2005). Most data showed a deuterium excess of <10 ‰ (Fig. A3), which was

attributed to an evaporative fractionation signal and enriched precipitation in summer (Ala-aho et al., 2018). The hinterland component for control areas and drained outside areas was an important source for water input in this context, slowly supplying water from adjacent connected areas (O'Connor et al., 2019). Therefore, the relative location of a site within a larger area with multiple topographic features, is pivotal for water accumulation or discharge. The drained sites showed first an increase in stable water isotopic composition and decreased at the end of the study period. This trend can be associated with an increase

of the permafrost thaw signal (Fig. 6, Fig. 7) because early summer water sources (snow melt and precipitation) have largely drained out. Furthermore, Ala-aho et al. (2018) highlighted that snow melt water contributed much more to summertime water flow than expected. This is possible when water that was initially replenished in local depressions can interact with suprapermafrost groundwater when flow regimes are lower during the mid-July low-flow (Ala-aho et al., 2018). Surface water isotope signals gradually increased and were the most influenced by precipitation. Including the full isotopic dataset from

2016–2019 in this study, allowed a more general view on monthly data variability. The largest difference in isotopic data was found for surface waters from June to July indicating that in June waters substantially consist of snow melt, while already in July waters are dominated by precipitation (Fig. 6).

### 4.3 Drainage feedbacks on thaw depths dynamics

   Small-scale thaw depth variations influence the area regarding active layer movement and exchange between surface water

and suprapermafrost groundwater. Subsurface water flow affects permafrost due to shifts in thermal conductivity and heat capacity (Sjöberg et al., 2016; Walvoord and Kurylyk, 2016). Both factors influence the seasonal development of thaw depth in permafrost soils and are strongly determined by the hydrologic status. In the beginning of the growing season, very wet soils have a high heat capacity, which slows down the deepening of the thaw layer initially. Starting by mid of July, the wetter microsites have lost enough standing water to considerably lower the heat capacity, while thermal conductivity is high, so that

thaw progresses further and stronger into the fall (Fig. 5). Our observations demonstrate that drainage speeds up the initial drying of topsoil layers following the flooding in early summer. As a consequence, microsites affected by drainage quickly reach a status with relatively low heat capacity and still high enough thermal conductivity that promotes a quick progression of the thawing front. However, with the drying out of organic topsoil layers quickly progressing already in June, the decreasing thermal conductivity dominates the process here, and mostly slowed down the thawing capabilities already by mid-July (Fig.

5). Thus, even though the uppermost layers heat up strongly with energy input in summer, this energy cannot be conducted





into deeper layers due to low thermal conductivity, which also slows down the thawing process. These effects were also shown in a previous study in the same study area (Kwon et al., 2016).

**4.4 Drainage impacts on site characteristics and biogeochemical cycles**

Vegetation adapted to high water levels, such as cotton grasses (*Eriophorum angustifolium*) and tussocks (*Carex* species),
developed at the predominantly wet areas (Fig. A2) within the undisturbed floodplain (Kwon et al., 2016). The change in hydrologic status also led to a long-term shift of the main vegetation type towards shrubs and tussock, with shrubs dominating the drained area (Goeckede et al., 2017; Kwon et al., 2016). Shrubbier vegetation was able to develop a deeper and larger root system when soils were drier. Drier and warmer topsoils generated by drainage promoted this change in vegetation. As a consequence, the energy balance (snow cover, shading) can be altered and evaporation may lead to further changes in the
annual hydrologic regime. Furthermore, such vegetation types require more water from the soil and help retain the soil wetness status. With more water uptake, vegetation was also able to enhance evapotranspiration (Chapin et al., 2000; Merbold et al., 2009). This may further dry out soils and promote vegetation with a deeper rooting zone, reinforcing and confirming the change towards drier soil conditions and an enhanced channeled flow (Liljedahl et al., 2016).

Overall, more water leaves the drained (inside and outside microsites) area with varying exchange with the organic soil layer
over the growing season. The increased groundwater discharge towards surface waters was consistent with findings of many other studies (Connon et al., 2014; Déry et al., 2009; Evans and Ge, 2017; Frampton et al., 2013; Kurylyk et al., 2014; Lamontagne-Hallé et al., 2018; Walvoord and Striegl, 2007). This leads to varying carbon production and transformation on site and transport towards surface waters (Walvoord and Striegl, 2007). The quicker suprapermafrost discharge within the drainage area could lead to more rapid lateral transport of constituents towards surface waters (Walvoord and Kurylyk, 2016).
The faster water flow during and directly following the spring freshet (June), and with higher precipitation inputs (September), is expected to laterally transport higher dissolved organic carbon (DOC), and these concentrations decrease towards the warmest time in summer (Guo et al., 2015; Prokushkin et al., 2009; Vonk et al., 2015). Therefore, lowest DOC concentrations were expected during the low-flow period in July and August in 2017. During the driest period in July, water transport and leaching of carbon were limited to the mineral soil layer. Instead, there is a stronger focus on collecting permafrost thaw water
within the mineral layer, but due to low hydraulic gradients and conductivity, exported water masses are relatively low. With low water tables creating drier soils, the potential for microbial respiration increases, which lead to a shifted $CO_2$ production (Goeckede et al., 2019; Kwon et al., 2019). Furthermore the birch effect, which represents a quick release in $CO_2$ due to soil rewetting (e.g., precipitation events), also leads to changes in carbon export in comparison to natural wet soil conditions (Singh et al., 2023). But $CH_4$ production in dry areas is much more reduced due to limited water saturation and anoxic conditions
(Bastviken et al., 2008; Dabrowski et al., 2020).





## 5 Conclusion

This study illustrated expected future conditions as a consequence of climate warming by simulating permafrost degradation with an artificially constructed drainage ditch in an Arctic floodplain underlain by permafrost. We compared water flow dynamics and stable water isotope signatures over the growing season in 2017. Our core findings were pronounced differences

in water levels and their temporal dynamics between control and drained area. Absolute water levels were higher at the control area due to widespread inundation. Lower water levels at the drained area led to stronger temporal fluctuations, stronger impact of precipitation events, drier soils and the development of higher vegetation (shrubs in contrast to cotton grass). Due to limited heat capacity and lower thermal conductivity in dry soils the thaw depths were generally shallower at the drained area.

The presence of a drainage ditch modified lateral transport conditions as well as the redistribution of suprapermafrost

groundwater. This led to different water transport mechanisms between the two areas. An increase in hydraulic gradients from larger microtopographic differences caused by the drainage ditch resulted in overall quicker lateral water flow velocities. The main flow direction was towards the drainage ring and towards the southeastern area, where the drainage ring converged and where the outlet of the drainage ring was located; from there the floodplain water flowed into the Ambolikha River. Slower control area water flow was affected by larger soil saturation zones (inundation, organic and deeper mineral soil layer due to

larger thaw depths), and was more influenced by the hinterland, where water discharged from the connected surrounding area. Transport direction depended on very small-scale shifts in water tables and followed belowground channels.

The stable water isotope differences between the two areas indicated the different roles of water sources. Depleted isotopic signatures in June were highly influenced by the spring freshet resulting from the snow melt. In the course of the growing season, isotopic signals followed a general precipitation signal, when the content of snow melt water decreased. At the end of

the season, differences between the drained and control area became more pronounced: with a limited groundwater buffer at the drained area, rain water was laterally discharged quicker off the floodplain and permafrost melt water was less diluted particularly in the late season. In control areas, a well-mixed suprapermafrost water body due to longer inundation periods reflected the isotopic signatures. The control area was vertically and laterally better connected over the region, where suprapermafrost groundwater and precipitation were discharged at a slower rate. Permafrost melt water isotopic signal was

more diluted over the whole water body, even though it was deeper thawed.

Such a unique study site demonstrated adequate requirements to compare and characterize natural conditions with a drainage influenced area regarding suprapermafrost water level shifts. The accelerated water flows, drier soil conditions, higher vegetation development and a stronger permafrost melt water input in the late season will potentially lead to a shifted carbon distribution over the area. It will also result in shifted carbon export, e.g., quicker lateral carbon transport into the surface

waters and ultimately into the Arctic Ocean, where the fate of carbon might experience various changes. At locations with drier soils, shrubbier vegetation with a larger root system is able to develop and can influence carbon uptake and soil respiration as a consequence of shifted soil water status. Future studies should focus on combining lateral and vertical water and carbon





fluxes to better understand and quantify the carbon processes and transport pathways in response to projected shifts in water flow dynamics in tundra ecosystems.

**6 Appendices**

**Appendix A**

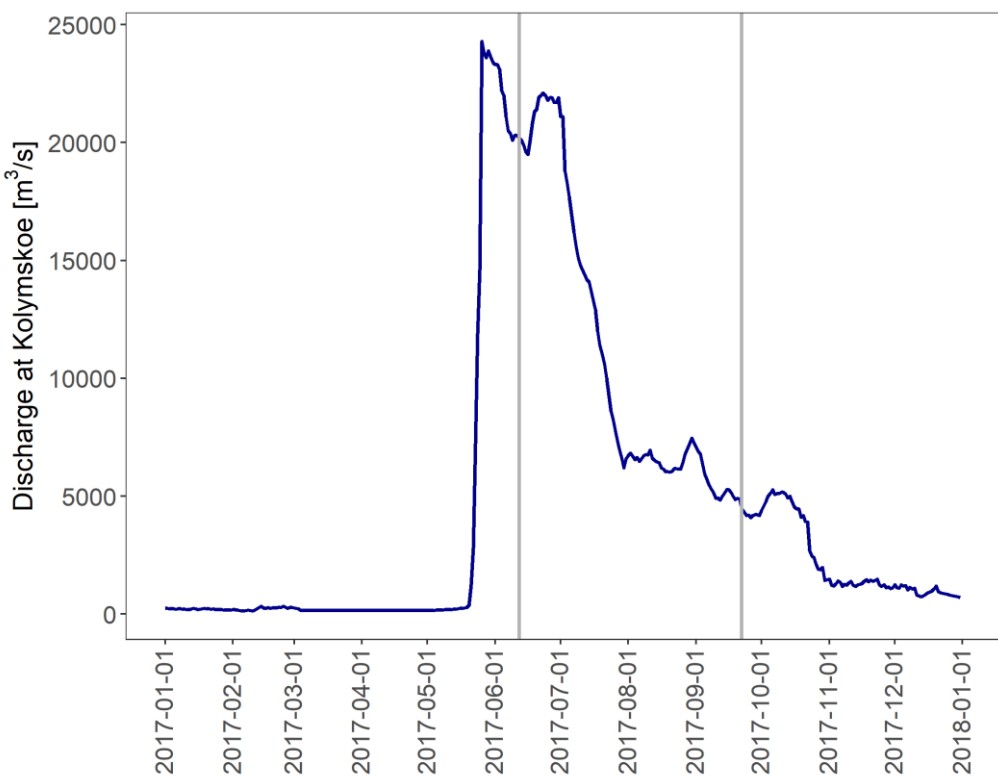

**Figure A 1: Hydrograph at Kolyma station Kolymskoe showing a nival streamflow regime. First peak during spring freshet was between 26 May 2017 and 03 June 2017; second peak was between 20 June 2017 and 01 July 2017.; summer low-flow was between 635 30 July to ca. 30 October 2017. Grey lines represent start (12 June 2017) and end (22 September 2017) of ultrasonic water level measurements. Data: McClelland et al. (2023)**





**Table A 1: Water type description and wetness indicator throughout the measurement period for each location.**

| Hydrological section | Description | Wetness indicator, WI [m] and location ID | |
|---|---|---|---|
| | | **dry**<br>WI <= −0.138 | **wet**<br>WI > −0.138 |
| **D-in** | drainage inside sites: all locations within the drainage ring | DI-01, DI-03, DI-04, DI-05, DI-07, DI-09 | DI-02, DI-06, DI-08, DI-10 |
| **D-out** | drainage outside sites: all locations adjacent to the drainage ring | DO-09 | DO-01, DO-02, DO-03, DO-04, DO-05, DO-06, DO-07, DO-08 |
| **Ctrl** | control sites: measurement at the control, non-manipulated site | C-04, C-05 | C-01, C-02, C-03, C-06, C-07, C-08, C-09, C-10 |
| **SW** | surface water sites: measurements at the drainage ditch | | SW-1, SW-2, SW-3 |


**Table A 2: Water level distance measurement outliers in 2017.**

| ID | Outliers |
|---|---|
| SW-2, DO-04, C-01, C-08 | none |
| C-03, DO-08, C-07, C-06, DO-07, DI-08, DO-01, SW-1, DO-06, DO-09, DI-10, C-09 | low (< 9 %) |
| C-02, DI-09, DI-03, C-04, DI-07, DI-04, C-10 | medium (10–40 %) |
| C-05, DI-02, DI-05, DO-02, DI-06, DO-03, DO-05, DI-01, SW-3 | high (> 40 %) |





**Table A 3: Four selected precipitation events throughout the measurement period.**

| Selected precipitation events | | | | | | | |
|---|---|---|---|---|---|---|---|
| **1** | | **2** | | **3** | | **4** | |
| 02–03 July | | 29 July | | 16–17 August | | 03 September | |
| Precipitation sum [mm] | | | | | | | |
| 4.5 | | 5.3 | | 5.3 | | 5.7 | |
| Precipitation duration [h] | | | | | | | |
| 4 | | 2.5 | | 4 | | 8 | |
| Precipitation intensity [mm h$^{-1}$] | | | | | | | |
| 1.1 | | 2.1 | | 1.3 | | 0.7 | |
| ID | Abs. WL change [m] | ID | Abs. WL change [m] | ID | Abs. WL change [m] | ID | Abs. WL change [m] |
| DI-03 | 0.003 | DI-03 | 0.002 | C-03 | 0 | C-09 | 0.003 |
| DO-08 | 0.003 | C-03 | 0.003 | DI-03 | 0.002 | C-01 | 0.004 |
| C-03 | 0.004 | DO-05 | 0.005 | DI-04 | 0.002 | DO-04 | 0.008 |
| C-08 | 0.004 | C-10 | 0.006 | C-08 | 0.005 | C-02 | 0.008 |
| DI-01 | 0.005 | C-01 | 0.007 | C-01 | 0.007 | C-03 | 0.008 |
| C-01 | 0.005 | C-09 | 0.009 | C-07 | 0.011 | DO-01 | 0.009 |
| C-02 | 0.009 | C-02 | 0.01 | C-10 | 0.012 | DO-08 | 0.009 |
| C-07 | 0.009 | C-07 | 0.01 | C-09 | 0.015 | C-07 | 0.009 |
| DO-01 | 0.01 | DO-01 | 0.013 | SW-1 | 0.02 | C-08 | 0.011 |
| C-10 | 0.01 | DO-03 | 0.013 | SW-2 | 0.02 | DO-07 | 0.012 |
| DO-03 | 0.011 | C-08 | 0.013 | DO-01 | 0.022 | DO-02 | 0.016 |
| DO-04 | 0.013 | DO-02 | 0.017 | DO-04 | 0.029 | DO-06 | 0.016 |
| DI-06 | 0.014 | SW-1 | 0.018 | C-04 | 0.037 | DI-05 | 0.02 |
| DO-02 | 0.018 | SW-2 | 0.018 | DO-08 | 0.04 | DI-08 | 0.025 |
| DO-06 | 0.018 | DO-04 | 0.019 | C-05 | 0.045 | C-06 | 0.027 |
| DO-07 | 0.02 | DO-07 | 0.021 | C-06 | 0.048 | SW-1 | 0.032 |
| C-09 | 0.022 | C-06 | 0.029 | DO-07 | 0.051 | SW-2 | 0.033 |
| SW-1 | 0.023 | DO-08 | 0.031 | DO-09 | 0.056 | DI-10 | 0.035 |
| DO-09 | 0.027 | C-04 | 0.037 | DO-06 | 0.078 | DI-04 | 0.037 |
| SW-2 | 0.027 | DO-06 | 0.051 | DI-08 | 0.08 | DI-07 | 0.037 |
| C-06 | 0.03 | DO-09 | 0.053 | DI-07 | 0.095 | DO-09 | 0.045 |
| DI-08 | 0.043 | DI-07 | 0.053 | DI-10 | 0.112 | C-04 | 0.051 |
| DI-04 | 0.045 | DI-08 | 0.054 | | | DI-09 | 0.059 |
| C-04 | 0.058 | C-05 | 0.054 | | | | |
| C-05 | 0.058 | DI-10 | 0.067 | | | | |
| DI-05 | 0.06 | DI-05 | 0.07 | | | | |
| DI-10 | 0.064 | DI-04 | 0.206 | | | | |
| DI-09 | 0.075 | | | | | | |
| DI-07 | 0.087 | | | | | | |





**Table A 4: Porosity measurement in 2018. On six locations across the study site, upper organic and lower mineral soil layers were sampled for porosity analysis (in total: 14 samples – seven of the organic and seven of the mineral layer). The mean transition between organic and mineral layer was 23 ± 3 cm below ground. Mean porosity values for organic material were 79 ± 4 % and for mineral material 24 ± 2 %.**

| ID | Sampling date and time | Transition between organic and mineral soil layer [cm] | Site type | Average thaw depth [cm] | Soil layer | Porosity [%] |
|----|------------------------|--------------------------------------------------------|-----------|-------------------------|------------|--------------|
| 1a | 08.07.2018 12:00 | 20 | D-in | 34.8 ± 1.3 | organic | 80.8 |
| 1b | 08.07.2018 12:30 | | | | mineral | 22.5 |
| 2a | 08.07.2018 13:00 | 26 | D-in | 38.0 ± 3.4 | organic | 84.8 |
| 2b | 08.07.2018 13:30 | | | | mineral | 24.8 |
| 3a | 08.07.2018 17:00 | | | | organic | 75.4 |
| 3b | 08.07.2018 17:30 | 23 | Ctrl | 43.6 ± 2.9 | mineral | 23.0 |
| 3c | 08.07.2018 17:45 | | | | mineral | 24.6 |
| 4a | 17.07.2018 14:00 | 22 | D-in | 34.3 ± 4.2 | organic | 81.7 |
| 4b | 17.07.2018 14:20 | | | | mineral | 27.8 |
| 5a | 17.07.2018 14:45 | 24 | D-in | 49.8 ± 1.7 | organic | 74.7 |
| 5b | 17.07.2018 15:05 | | | | mineral | 24.1 |
| 6a | 17.07.2018 15:40 | | | | organic | 77.1 |
| 6b | 17.07.2018 16:00 | 25 | Ctrl | 43.5 ± 2.9 | organic | 78.6 |
| 6c | 17.07.2018 16:20 | | | | mineral | 21.8 |




**Figure A 2: Vegetation distribution over drained (blue) and control area (orange). The predominant vegetation types at the drained (mainly dry) area are high shrubs and tussock with shrubs. The predominant vegetation types at the control (mainly wet) area are cotton grass and tussocks.**


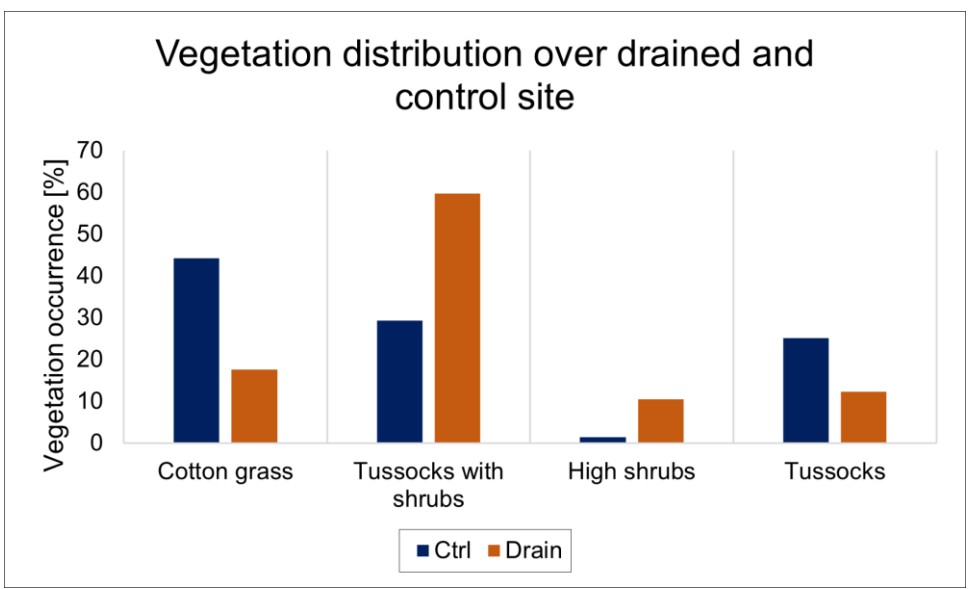





**Figure A 3: Time series of the water levels (blue line) combined with δ18O data (grey points) and D-excess (red points); at a wet location in the control area, C-01 (upper panel) and at a dry location within the drainage ring, DI-04 (lower panel).**




## Data availability

All raw data can be provided by the corresponding authors upon request.

## Author contribution

**Conceptualization**: S. Raab, M. Goeckede

**Data curation**: S. Raab

**Formal analysis and visualization**: S. Raab

**Funding acquisition**: M. Goeckede, J. Vonk

**Investigation**: S. Raab, M. Goeckede, A. Hildebrandt, K. Castro-Morales, J. Vonk, M. Heimann, N. Zimov

**Methodology**: S. Raab, M. Goeckede, A. Hildebrandt, K. Castro-Morales

**Project administration:** S. Raab, M. Goeckede

**Resources**: S. Raab, M. Goeckede, A. Hildebrandt, K. Castro-Morales, J. Vonk, M. Heimann, N. Zimov

**Writing – original draft:** S. Raab, M. Goeckede

**Writing – review & editing:** S. Raab, M. Goeckede, A. Hildebrandt, K. Castro-Morales, J. Vonk, M. Heimann, N. Zimov

## Competing interest

The authors declare that they have no conflict of interest.

## Acknowledgements

This research was supported by the European Commission Horizon 2020 framework programme Nunataryuk (grant no. 773421). Further funding was provided by the European Research Council (ERC) under the European Union's Horizon 2020 research and innovation programme (grant agreement no. 951288, project Q-Arctic). The project was further supported by the International Max Planck Research School for global biogeochemical cycles (IMPRS-gBGC) and the Max Planck Institute for Biogeochemistry (MPI-BGC) in Jena, Germany. The authors thank the Field experiments and instrumentation service group at MPI-BGC for constructing the piezometers and the technical support on site (esp. Olaf Kolle, Martin Hertel and Martin Kunz). We also appreciate the help of the staff members of the Northeast Scientific Station (NESS) in Chersky for facilitating field and laboratory experiments (esp. Wladimir Tataev, Galina Zimova, Anna Davidova). Field work was supported by Linus Schauer, Megan Behnke, Martijn Pallandt and Kirsi Keskitalo and water stable isotope measurements were conducted at MPI BGC-IsoLab (Heiko Moosen and Heike Geilmann). The authors thank Judith Vogt and Christoph Raab for internal reviews.

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
