# Peer review of "Small-scale hydrological patterns in a Siberian permafrost ecosystem affected by drainage"

_Biogeosciences, 2023_

## Author Response (AR1)

Referee 1:

To structure the answers for the reviewer feedback, we added the referee comments in italic and the answers in normal text style with blue coloring. The manuscript will be edited according to the changes as described in our answers below.

*General comments:*

*The topic of this paper may be of interest to the journal but currently it is not well written or presented. There are lots of short paragraphs and poor punctuation. I recommend someone proof read this to save reviewer's time and patience. The discussion could be reduced substantially by removing the areas that are simply a rehashing of content that was already presented. This Is particularly the case for the first few sections. The Conclusion needs a complete rewrite as it is also mostly just a repeat of what ahs already been said. Why does this study matter? What are the ramifications? How can these results help project the future state of permafrost and hydrologic changes? These are areas that this paper can address yet they are barely mentioned. After reading this I am left wondering why do we care?*

Thank you for your feedback on the discussion paper *Small-scale hydrological patterns in a Siberian permafrost ecosystem affected by drainage by Raab et al.* We will work on changing the structure as outlined above, with detailed changes described also further below in our answers to the comments with line numbers. We will also emphasize the rationale and importance of the study and its implications/ramifications. After working in all these changes, we will have a professional language editor proof-read the final version of this manuscript. We think that these changes addressing your comments will substantially improve the paper.

*Comments keyed to the line numbers:*

*19: by "ice table" do you mean "top of near-surface permafrost"?*

Exactly. We will change it to: "above the permafrost table"

*23: centers at or is focused at*

We will change it to: "is focused on"

*23: the drainage ditch reflects…*

We will change it to: "drainage ditch was constructed in 2004 to simulate…"

*27: Over or through the drier areas? "In" does not seem correct here.*

We will change it to: "water flow through drier areas"

*quicker water flow in drier areas where/how? Through these areas?*

*"iii) larger saturation zones in wet areas" at the drained site? I thought it was drained.*

iii) We looked at a drained and control (mostly wet) area. Both areas are characterized by wet and dry parts, but there are more wet areas at the control area and relatively drier sections at the drained area. This sentence generalizes: saturation zones are larger at wet parts of the whole investigation area in general. We will change this statement to: "iii) larger saturation zones in wetter areas

*29: "Shifts in hydraulic connectivity associated with a shift in vegetation abundance and water sources may impact carbon sources, sinks as well as its transport pathways" Before this vegetation has not been measured as a control or an aspect of the study.*

Thank you for highlighting this. We will change this sentence to: "Shifts in hydraulic connectivity in combination with a shift…" to highlight, that the effects appear simultaneously. In former studies (e.g. Corradi et al. 2005), vegetation structure within the drainage area before implementation of the drainage has been analyzed. It was found, that the drainage area was also covered by cotton grass and tussocks, with a very similar composition as later also observed within the control area. Later studies (Kwon et al. 2016) identified vegetation shifts towards shrubbier plants and a higher abundance of tussocks instead of cotton grasses as a consequence of the drainage.

*34: Note some studies say 24%.*

Thank you for pointing this out. Obu (2021) distinguished between the 'permafrost region' and the 'permafrost area': the 'permafrost region' is the combination of all permafrost zones (continuous, discontinuous, sporadic etc.), whereas the 'permafrost area' only accounts for the area actually underlain by permafrost. The 'permafrost area' is accordingly smaller than the 'permafrost zone', and therefore the 15 % value, referring to the permafrost area also is much lower than the e.g. 24 % referring to the permafrost zone that can be found in other literature.

*44: "the permafrost soil profile" maybe say "top section above permafrost soils"*

We will change it according to your suggestion: "top section above permafrost soils"

*46: present references in chronological order unless the journal says otherwise. And please correct me if I am wrong for this journal!*

Thank you for having this in mind. We looked into the guidelines: "In terms of in-text citations, the order can be based on relevance, as well as chronological or alphabetical listing, depending on the author's preference." We've chosen the alphabetical order.

*47: forming an aquiclude at the base of the active layer*

We will change it accordingly: "forming an aquiclude at the base of the active layer"

*57: "Arctic mineral soils" maybe more like ice cemented silt or silty soils in general?*

*Exactly. We will highlight this with: "Arctic mineral (silty) soils"*

*65-69: can you provide a way to end this paragraph and lead into the Siberian floodplain one without such a jarring change in topic?*

Thank you for pointing this out. We will change it to: "…season.

Seasonal soil water conditions are characteristic of Siberian floodplains. These areas are affected…"

*83: "and small (e.g., Quinton et al., 2000; Walvoord and Kurylyk, 2016) scales"*

We will change it accordingly: "and small (e.g., Quinton et al., 2000; Walvoord and Kurylyk, 2016) scales"

*85-89: Same as earlier comment- can you establish a summary/reason to move more seamlessly from 85-87 to "In this study"?*

We will change it to: "…future research. To address the shifts in potential carbon distribution, we first need to understand patterns of water table and thaw depth.

Hence, we investigated…"

*113: provide a month/timeframe for the end of the growing season.*

*Same for "spring"*

We will change it to: "growing season (October). […] spring (May - June)"

*123-125: Can these two sentences be integrated elsewhere so as to not have this short paragraph?*

We will integrate this paragraph to the previous paragraph.

*131-133: can this be integrated into the previous paragraph?*

We will integrate this paragraph to the previous paragraph.

*137: "refer to Kittler et al., (2016)"*

We will change it to: "see Kittler et al. (2016)"

*135-139: can this be one paragraph?*

We will move this paragraph to the previous one with the transition phrase: "In this study, we analyzed…"

*150 and elsewhere: provide company location/address*

In 150: We will change it to: "(MB7380 HRXL-MaxSonar-WRF, MaxBotix, Brainerd, Minnesota, USA)"

And 232: "(YSI Inc., Yellow Springs, Ohio, USA)."

And 233: "filtered (0.7 μm GF/F Whatman®, VWR International GmbH, Darmstadt, Germany)"

And 236: "(Finnigan MAT, Bremen, Germany)"

*160: what/how much were the measurement errors?*

Here, we addressed measurements that were excluded from further analysis, since the errors were too large (up to several cm) due to water droplets in the pipe, e.g. But we included the general measurement resolution in line 151: "The measurement range of the sensor is 0.3 to 5 m and the measurement resolution is 5 mm (MaxBotix Inc., 2023a)."

*198: Bouwer and Rice, (1976)*

We will change it to: "Bouwer and Rice (1976)"

*203: provide a sentence about the composition of the mineral layer*

In lines 124-125, we have already included this information: "The underlying silty-clayey mineral layer originated from river and flood water transport."

*228: "This additional data resulted from own measurements on-site in 2016 and 2018"?*

We will change it to: "This additional data resulted from our on-site measurements in 2016 and 2018"

*296-299: is there data to back this up? Can you say anything about what are clearly high hydraulic conductivities of the soils if this is the case. Perhaps there is a seasonality to this response that changes as the active layer expands downward? I have never seen this reported before.*

The described feature is visible in most of our own measurement data, but we did not consider it a core result for the objectives of the study presented in this manuscript, and therefore did not show any details. In the revised version of the manuscript, we will add a section to the appendix that features these datasets, showing representative diurnal cycles of suprapermafrost groundwater levels, and how they evolve over the course of the growing season.

We will link this information towards the appendix by: "These fluctuations can be observed for all sites but were most pronounced at drainage-inside sites (Appendix Figure A 4)."

*315: maintained may be better than retained*

We will change it accordingly to "maintained"

*326: and a maximum of 0.88L….*

We will change it accordingly to: „and a maximum of 0.88 L…"

*357: perhaps say "A transition zone" unless you mean the active layer transient layer? If you just mean there is a transition zone between soil horizons perhaps state it that way to avoid confusion.*

We will change it to: "Regarding the vertical structure of the soil profile, a transition zone…"

*373: any idea on the age of the permafrost ice and the climatological regime under which it formed?*

We do not have radiocarbon data of permafrost ice, so we cannot date the age of this source. However, it is important to note here that ice wedges and ground ice are more depleted than rain water, and therefore we can clearly distinguish between these two sources.

Opel et al. (2011, https://doi.org/10.1002/ppp.667) analyzed Siberian ice wedge stable water isotopes in combination with radiocarbon data. They found that stable isotopes were between -22 to -25 ‰ ($\delta^{18}O$) and −168 and -194 ($\delta D$), which is in line with our (top layer) permafrost ice samples ($\delta^{18}O$ = 23 ‰, $\delta D$ = 181 ‰). For the radiocarbon, they found youngest ages ranging between 470 to 832 a BP and oldest data up to 10228 a BP, depending on the ice wedge itself. Given the similar water isotopic signatures, it is likely that our permafrost ice was formed in the Holocene under colder climatic conditions (e.g. Little Ice Age).

To point this out, we will add the sentence: "…. Welp et al. (2005). Permafrost ice data were also in line with stable water isotopes analyzed by Opel et al. (2011), who were investigating radiocarbon in Siberian ice wedges. Linking water isotopes with radiocarbon data, we can assume that our permafrost ice was formed in the Holocene (Little Ice Age)."

*Maybe add arrows on the chart pointing to where the snow and rain values would be and provide the numbers on the chart? I suspect you do not want to expand the chart to add their true data points as that will push your data too close? But providing the values on the chart would be good.*

Exactly. If we would include all rain and permafrost ice data, it would push the data too close. We will add the mean ± sd data for rain and permafrost ice in the chart as well as arrows pointing into the rain and permafrost depletion/enrichment direction.

*376: of wet or of summer precipitation? Snow is also precipitation.*

*Are you saying the permafrost melt water is snow melt water? It more likely has a strong summer precipitation component as the ground is/was frozen when the snowmelt occurs.*

Thank you for pointing this out. It was not written precise enough. We will change it to: "precipitation during the growing season" to address rain precipitation.

Permafrost melt and snow melt water are not the same, but both show more depleted isotopic signatures compared to rain and summertime groundwater and surface water isotopes. Most groundwater samples were more depleted in spring (snow melt/spring freshet influence during first samplings) and became again more depleted towards the end of measurements due to permafrost melt water input (depletion esp. at the drained area).

We further will change it to: "…58 ± 9 % of snow (spring samples) and permafrost melt water (late summer and autumn samples, Fig. 7)."

*Figure 7 caption. Do you mean "of the samples"?*

We will change it accordingly to: "of the samples"

*389: a nearby control*

We will change it accordingly to: "a nearby control area"

*399:areas?*

*"could be observed" is probably ok but earlier "were" is used. So perhaps use "was"?*

We will change it to: "throughout the region"

We will change it to: "was observed"

*394-409 is a lot of repetition of what has been said at least once and in some cases twice already.*

394-409: We will revise this paragraph, and remove repetitions to previous statements.

*420" generally higher*

We will change it accordingly to: "generally higher"

*424: rainfall, higher evaporation rates, combined*

We will change it to: "rainfall, higher evaporation rates, combined"

*543: no comma needed after features*

We will remove the comma.

*547: This is possible when water that was initially replenished in local depressions can interact with*

*Check tense here and throughout.*

We will change it to: "This is possible when water that was initially replenished in local depressions interacted with suprapermafrost groundwater when flow regimes were lower during the mid-July low-flow."

*Conclusion section is again mostly a rehash of what has said at this point three times. The real conclusions start around line 621. The paper could be strengthened quite a bit by weaving in more "why do we care/impact" statements. The random "carbon" is probably not needed. The real gems of this study are ramifications for Infrastructure, thaw slumps, and thermo-erosion gully development- ways the results of this study can help project how future warming will affect permafrost thaw morphologies.*

*Other topics of relevance are climate impacts on surface hydrology, as well as how, where, and when wildfire or other disturbances that lead to subsidence, ice melt, and thermokarst will alter hydrology. The small excavated channel can be seen as representing what permafrost will do in hotspots in a warmer future world.*

We agree with the reviewer that repetitions should be avoided. Therefore, the manuscript text will be checked carefully again, and particularly the conclusions will be edited accordingly.

Regarding conclusions with respect to infrastructure, the authors are rather hesitant to link their findings to impacts in the fields highlighted by the reviewer. We certainly agree that infrastructure failure and changes in surface structure of ecosystems following the degradation of permafrost soils are closely connected to hydrological processes. Still, our own study worked with a man-made drainage as disturbance, and studied the hydrological responses that followed it. Accordingly, there is no connection between permafrost status and the initialization of thermo-erosion, and we also cannot draw conclusions how the hydrological status affected the geomorphology. But we will highlight that there are different kinds of disturbances (e.g. wildfires) that can lead to subsidence and changes in hydrological conditions (e.g. drainage). And we will link this to climate impacts that can lead to the scenario – intensified drainage – that we studied.

Referee 2:

To structure the answers for the reviewer feedback, we added the referee comments in italic and the answers in normal text style with blue coloring. The manuscript will be edited according to the changes as described in our answers below.

*This is an important study that investigates supra-permafrost water table depths, water flow velocities, and vertical and lateral transport and connectivity in an Arctic floodplain under degrading continuous permafrost conditions. The experimental setup provides a great opportunity to simulate drainage associated with permafrost degradation and test its impacts on supra-permafrost hydrology and active layer thaw depth. The authors discuss how the changes to the supra-permafrost groundwater conditions can interact with thaw depth, vegetation composition, and carbon cycling and transport, which has important implications regarding ecosystem transition/succession and landscape carbon budgets amid continuing permafrost degradation. The paper is generally well written (specific line comments listed below). The prefixes "C", "DO", and "DI" are helpful when reading through the text to determine which type of sample is being discussed (e.g., "DI-08"). However, with these prefixes, it is not always clear if a particular sample being discussed is from a 'wet' versus 'dry' area (for example, Figure 5 differentiates between wet and dry areas).*

Thank you for your feedback on the discussion paper *Small-scale hydrological patterns in a Siberian permafrost ecosystem affected by drainage by Raab et al.* We will change the manuscript according to the opinions and suggestions of both reviewers, with detailed edits listed further below.

Regarding the general comment addressing ID names: The differentiation of sample ID's regarding the area (drainage inside area, control area etc.) seemed more important for all subsequent analysis. Even though there are dry sites at the control area, this area showed wetter conditions in general. And there are wet sites at the drained area, but this area showed drier conditions in general. We included Table A 1 in the appendix for information about the different wetness status for each site and hope that this is sufficient enough and makes e.g. Fig. 5 better readable.

*Discussion section 4.2 discusses microtopographic features. How was microtopography calculated? For example, were local elevations and local depressions determined based on the entire site area (control and drainage areas), or determined locally? For the second option, the microtopography could be calculated relative to a moving window median (or moving window mean, etc). I am also interested to know what the size distribution is for the local elevations and depressions, in terms of their horizontal area and vertical range. It would also be helpful for readers to see a map of the elevation that was used to calculate microtopography. For example, a high-resolution DEM was described. It would be helpful to show the DEM map, if available, and describe any large-scale gradient in the elevation across the entire site area.*

The microtopography was visually analyzed with the help of drone data and ground truth data. The drone data gives us high-resolution pictures throughout both areas but it only represents the top of vegetation and the land surface elevation could not be calculated. We will include the results of this drone picture in the appendix to better understand that higher elevated areas showed drier soil conditions and shrubbier vegetation. Therefore, we can qualitatively estimate the difference between drier and wetter areas. The small ridge (see comment below regarding the dry hummock) is visually detectable.

In the text, we will link this to the appendix: "In order to determine the absolute water level above sea level, we obtained the elevation of each of the monitoring locations across the Ambolikha site based on a 2018 drone survey that produced high-resolution digital elevation maps (Fig. A 5)"

*Also related to the microtopography, line 503 describes a dry hummock. Is the hummock identified based on its local microtopographic elevation? It seems that there are different uses of the term 'hummock' in the literature, e.g., sometimes referring to small mounds of Sphagnum moss. Is microtopography at this site driven by vegetation growth, or perhaps permafrost structure or other process? This might impact vegetation and lateral flows amid continued warming and permafrost degradation.*

The area referred to as 'hummock' in this section is actually a slightly elevated ridge with horizontal dimensions of several 10s of meters. To avoid misunderstandings, we will change the description of this landscape feature to 'small ridge': "…a small dry ridge was identified in the area (C-04, C-05 and C-06). South of this ridge the water flow direction…"

Although we do not know the long-term development history of this site, it is likely that this ridge was formed by preferential sedimentation during repeated flooding events following snow melt. It is possible that the trigger of such localized sedimentation is driven by vegetation, but this could only be speculated upon.

*Specific line comments:*

*Line 34: "approximately 15 % of the land surface" (extra space between 15 and %).*

There is already one space between 15 and %.

*Line 37: "organic-rich loess soils", perhaps include term yedoma for readers specifically interested in yedoma deposits?*

We will change it to: "organic-rich loess Yedoma soils"

*Line 228-229: "additional data resulted from own measurements", not sure if there is a word missing; perhaps use 'our' instead of 'own'.*

We will change it to: "additional data resulted from our on-site measurements"

*Line 260: It is not clear what is meant by "the largest fractions of the drainage area"*

We will change it to: "most parts of the drainage area"

*Line 310: Figure 4: It looks like some of the sample ID numbers are labeled for some dates and not others. For example, SW-3 appears in the drained area for the 15 June and 15 September figures, but not for the 15 July or 15 August figures. Is this due to limited space for labels? Or was the SW-3 site not sampled in July and August?*

*Exactly, some ID's were not displayed due to limited space (resulting from lines and arrows), but all data points were used. Thank you for pointing this out. We will change the figure, so that all labels will be visible.*

*Line 349: Where it says "Low thaw depth tables", it's not clear if this refers to deeper thaw or thaw closer to the surface.*

We will change it to: "Shallow thaw depth tables"

*Line 353: "…where the increase was mainly resulted…", perhaps "was" can be removed.*

We will remove "was": "…where the increase mainly resulted…"

*Line 405: "where precipitation was absent and evaporation rates high due to higher air temperature", it sounds as though this is discussing variation in time, rather than variation in space? If this is the case, perhaps use "when" instead of "where."*

We will change "where" to "when" as suggested.

*Line 406: Similar comment about "where" vs "when"*

We will change "where" to "when" as suggested.

*Line 420: What defines a long-term precipitation event?*

We did not define this term in particular. We refer here to the continuous precipitation period starting in mid of September (Fig. 3). Therefore, we will change the text to: "During long-term precipitation events (mid-September precipitation period, Fig. 3) this effect…"

*Line 420: Perhaps use 'generally' rather than 'general'*

We will change "general" to "generally".

*Line 439-440: The sentence reads, "At dry sites, transport was limited to the organic soil layer without inundation", but it's not clear if this means "transport only occurred in the organic soil layer without inundation"* **or** *"transport into the organic soil layer was limited when it was not inundated".*

We will change the statement to clarify the meaning: "At dry sites, transport was limited to the organic soil layer where inundation was missing and water flow less vertically…"

*Line 575: The sentence reads "vegetation types require more water from the soil and help retain the soil wetness status." It sounds like these are two contrasting processes. Is the main idea that the net result will be drier soil conditions?*

With "help retain the soil wetness status" it was meant, that the specific wetness (which is comparably dry at the mentioned area) is maintained. We agree that this sentence might be misleading, so we will change it into: "help retain the reduced soil wetness status"